# Intra-Day Solar Power Forecasting Strategy for Managing Virtual Power Plants

**DOI:** 10.3390/s21165648

**Published:** 2021-08-22

**Authors:** Guillermo Moreno, Carlos Santos, Pedro Martín, Francisco Javier Rodríguez, Rafael Peña, Branislav Vuksanovic

**Affiliations:** 1Department of Electronics, University of Alcalá, Alcalá de Henares, 28805 Madrid, Spain; guillermo.morenob@uah.es (G.M.); pedro.martin@uah.es (P.M.); 2Department of Signal Theory and Communications, University of Alcalá, Alcalá de Henares, 28805 Madrid, Spain; carlos.santos@uah.es (C.S.); rafael.pena@uah.es (R.P.); 3School of Engineering, University of Portsmouth, Winston Churchill Ave., Portsmouth PO1 3HJ, UK; branislav.vuksanovic@port.ac.uk

**Keywords:** power forecasting, long short-term memory recurrent neural network (LSTM-RNN), virtual power plant (VPP)

## Abstract

Solar energy penetration has been on the rise worldwide during the past decade, attracting a growing interest in solar power forecasting over short time horizons. The increasing integration of these resources without accurate power forecasts hinders the grid operation and discourages the use of this renewable resource. To overcome this problem, Virtual Power Plants (VPPs) provide a solution to centralize the management of several installations to minimize the forecasting error. This paper introduces a method to efficiently produce intra-day accurate Photovoltaic (PV) power forecasts at different locations, by using free and available information. Prediction intervals, which are based on the Mean Absolute Error (MAE), account for the forecast uncertainty which provides additional information about the VPP node power generation. The performance of the forecasting strategy has been verified against the power generated by a real PV installation, and a set of ground-based meteorological stations in geographical proximity have been used to emulate a VPP. The forecasting approach is based on a Long Short-Term Memory (LSTM) network and shows similar errors to those obtained with other deep learning methods published in the literature, offering a MAE performance of 44.19 W/m^2^ under different lead times and launch times. By applying this technique to 8 VPP nodes, the global error is reduced by 12.37% in terms of the MAE, showing huge potential in this environment.

## 1. Introduction

Around the world, the full deployment of solar energy is being facilitated by several factors including, but not limited to, the reduced price of solar panels; environmental, political and social concerns; and solar energy undercutting utility prices, inter alia. According to [1] global installed capacity will double every two years; however, significant factors have been identified which impede the speed at which solar dominance can be achieved: (i) lack of investments in efficiency, (ii) insufficient government incentives, and (iii) regulatory constraints. Small-scale Photovoltaic (PV) installations such those in the residential sector benefit from self-consumption by shifting a load from hours when electricity prices are high to hours when the PV energy is being generated, thereby achieving electricity bill savings. Going one step further, the aggregation and coordination of several PV installations in the shape of a Virtual Power Plant (VPP) with the accurate forecasting of global production facilitates its integration into the network [2]. Consequently, the increasing PV penetration can lead to the increasing aggregation of PV systems into VPPs. However, these new business models are difficult to implement due to the previously mentioned regulatory constraints.

Power forecasting along with load demand and energy prices, for different time horizons and resolutions, are factored into the equation. For VPPs, spatial horizons should also be considered. Forecasting methods can be classified according to different factors, such as: the forecasted parameter (irradiance or power), the time horizon and resolution, the lead time, the model approach, and the nature of the forecasting statistic. Regarding the forecasted parameter, two different alternatives exist: direct [3] and indirect [4]. The direct method predicts the solar power through historical datasets of PV power generation and weather conditions. Indirect forecasting differs from the direct method in that it firstly predicts the solar irradiance and then, the solar power is calculated by using a performance model of the PV plant. As far as the time horizon is concerned, four categories can be found [5,6]: nowcasting (from 1 min to several minutes) which is used for real-time optimization in Energy Management Systems (EMSs); short-term forecast (from 1 h to several hours) used for intra-day market participation and for day-ahead operation optimization; medium term forecast (from 1 month to 1 year); and long-term forecast (up to several years). Time resolutions may range from 1 minute for real-time market operations, and 15-minute periods for load-shifting strategies and for optimizing Battery Energy Storage Systems (BESSs), to 1 hour for longer time horizons used by consumption monitoring, and a 1-week resolution for 1-year time horizons which can be used to identify consumption trends [7]. The lead time can be defined as the time difference between the instant when the forecast is launched and the occurrence of the forecasted value, considering the forecast horizon as the maximum forecast lead time. Forecast errors increase with forecast lead time due to the atmospheric motion. As for the model, the optimal method for solar irradiance prediction depends on the forecast lead time [8]. In this regard, four approaches have been widely used [9]: (a) time-series-based statistical models whose aim is to identify patterns between historical datasets and the output parameters; (b) machine learning (ML) models mainly based on artificial neural networks (ANNs), which use historical datasets to learn the dependency between the past and the future; (c) physical strategies which utilize Numerical Weather Prediction (NWP) and PV models for solar power forecasting; and (d) hybrid models which explore different algorithm combinations with the aim of improving forecast accuracy and reducing the computational burden of online forecasting applications [5]. The objective of all the models is to improve forecasting accuracy by minimizing some quality metrics, usually the sum of squared errors. The existence of different models raises the question of whether one method is better than the others. This is particularly true for statistical and ML models. Some studies conclude that statistical models outperform ML models [10] while others state the opposite [11,12]. However, this interpretation may appear to be fairly simplistic without taking into account the dataset size [13], the variable being forecast [14], the time horizon [15], or the computational load [16]. Although historically, the forecasts have been dominated by statistical methods, over the last decade there has been a significant shift toward ML strategies [17]. This comparative study is beyond the scope of the paper. 

Regardless of the method used, the existence of forecasting errors poses a major challenge in optimizing the PV plant operation. While minor forecasting errors may not adversely affect the PV plant operation, larger errors can produce negative effects in the optimization models. Uncertainties hinder the performance in terms of accurately assessing the variables during the PV plant scheduling and operation. Forecast uncertainty quantification is, therefore, crucial. For this reason, considering the prediction intervals, which account for the uncertainty, provides additional accurate information about the expected values in terms of the range of plausible values and the probability assigned to each of them [17,18]. Another solution to the problem involves the aggregation of several PV sites for a unique forecasting strategy, since the error is significantly reduced as the number of installations increases. To prove this, in [19] the authors present an approach to forecast the PV power from irradiance prediction maps, obtaining the power forecast of 200 sites located in Germany. Results show that the error is reduced from a Root Mean Square Error (RMSE) of $0.11\;\mathrm{kW}/{\mathrm{kW}}_{\mathrm{peak}}$ for single sites, to $0.06\;\mathrm{kW}/{\mathrm{kW}}_{\mathrm{peak}}$ for an area of 220 km × 220 km with multiple sites. The distance among sites is also an important factor which influences accuracy, since the error is significantly reduced when the distance between facilities increases. This strategy provides a powerful solution in the context of VPPs, since multiple systems or nodes are controlled, managing Distributed Generation (DG) units, Energy Storage Systems (ESSs), flexible loads and Information and Communication Technologies (ICTs) [20]. Regarding the types of DG units, PV systems can be considered as the easiest and most cost-effective Renewable Energy Sources (RESs) to exploit, mainly for households, where it is possible to turn PV installations into flexible VPP nodes [21].

Finally, as stated above, for indirect forecasting approaches, performance models of PV systems are required to obtain the prediction of solar power generation. To this end, a strategy that works under arbitrary conditions of irradiance and temperature must be adopted. Methods that exhibit these key characteristics are the Osterwald’s method [22], which stands out by its simplicity, or similar studies from the literature that improve the performance of the Osterwald’s method by adjusting the results under low irradiance levels [23,24]. When the operating point of the PV panels is known, alternative methods, such as those reported in [25,26], can improve accuracy, while other research uses parametrization models to simplify the process [27]. Sometimes, the irradiance of the site is measured on a horizontal plane, obtaining the Global Horizontal Irradiance (GHI). However, the panels are on a different plane. This is typical in satellite measurements but can also be the case in installations with multiple Maximum Power Point Trackers (MPPT) or PV panels with axis trackers. To solve this problem, a conversion process is needed, using: (i) different expressions to tackle the problem step-by-step by separating the global components into direct irradiance, diffuse irradiance, and albedo, modifying the angle of these components to obtain the global irradiance on the plane of the panel, estimating its losses to obtain the effective irradiance, or (ii) an approach that simplifies the process [28]. In this regard, it becomes crucial to reduce the complexity and the computational burden placed on the forecasting algorithms. With this in mind, this work makes use of the Osterwald’s method to calculate the PV power, since low irradiance values (G<125 W/m^2^) are barely existent in the dataset and a generalization of the algorithm for VPP environments leads to better results. Satellite data are also required in this work since they offer information on the GHI, which is converted into irradiance on the tilted plane by following the steps stated above. 

The forecasting strategy developed in this paper, uses long short-term memory recurrent neural networks (LSTM-RNNs) and is based on an indirect approach in which the irradiance is forecasted first and the output power is calculated by using the PV model. LSTM-RNNs have been used in several works, achieving satisfactory results on account of their recurrent architecture, which includes memory units [16]. These allow the ANN to identify temporal patterns from the historical data of the forecast variable, thereby reducing the forecast error in comparison to other alternatives. The authors in [29] propose a PV power forecasting strategy based on LSTM-RNN which is compared with other methods without memory units, showing their limitations in terms of not being able to model the dynamics of the PV output power data. In [30] a LSTM-RNN with only exogenous inputs, e.g., dry bulb and wet bulb temperatures, and relative humidity, is used to forecast the day-ahead solar irradiance. 

The main contributions of this paper are summarized as follows: (i) the PV forecasting method is applied to a VPP environment to reduce the forecasting error, which is modelled as a function of two well-defined parameters called lead time and launch time; (ii) prediction intervals are used to model the forecast uncertainty as a function of not only the lead time and the launch time, but also the Cloud Cover Factor (CCF), which allows the type of day to be identified; (iii) the input data for the forecasting strategy are derived from free-of-charge open-access data sources, offering a viable and cost-effective solution; and (iv) a trade-off between accuracy and computational burden facilitates the application of multiple PV power forecasts at different locations, within the context of a VPP.

The remainder of this paper is organized as follows: Section 2 introduces the framework for the intra-day power forecasting strategy; the experimental results are presented in Section 3; and finally, some conclusions are drawn in Section 4.

## 2. Intra-Day Power Forecasting Framework

The proposed intra-day power forecasting strategy is depicted in Figure 1. It consists of four main blocks, namely: (i) input data; (ii) data preprocessing; (iii) model design and forecasting; and (iv) VPP coordination. The input data, which come from different sources, are fed to the preprocessing stage. The preprocessing step prepares the data as required by the training and forecasting models. Finally, the output of the forecasting algorithms is used as the input of the EMS of the VPP. In the following, the different parts are explained in detail.

### 2.1. Input Data

The input data consist of three specific categories according to the source and type of the information provided. The first category includes cloudiness and temperature, which are obtained from forecast maps, at different spatial and temporal scales, generated and regularly published by the Spanish agency of meteorology AEMET, via NWP [31]. The cloudiness dataset is used to define the Cloud Cover Factor (CCF), which indicates to what extent a cloud area on the NWP-based cloudiness maps creates shadows on the PV installation. This parameter is used to define the type of day: sunny, cloudy, and overcast. This allows the dataset to be split in different groups to create prediction intervals. Temperature data, on the other hand, are used to estimate the cell temperature of the solar panel at the prediction instant [32]. NWP-based weather maps are of great interest since some useful weather variables might not be available in solar installations. The deviation in the estimation of the cell temperature is then assessed by using the data obtained from the experimental setup, which is located at the Polytechnic School of the University of Alcala (Spain) and consists of a $2.97\;{\mathrm{kW}}_{p}$ PV facility with a meteorological station that gathers information of GHI, temperature and cell temperature [33]. The dataset, obtained from the PV facility, is taken during the period between 1 June 2020 and 31 May 2021, with a resolution of 15 min. In the second category, the Global Horizontal Irradiance (GHI) measurements are obtained from two sources: (i) a pyrometer, which is installed in the experimental setup and 30-second GHI measurements are taken and stored on the cloud (ThingSpeak) [34]; and (ii) the Copernicus Atmosphere Monitoring Service (CAMS), which provides a free historical dataset of the incoming surface solar irradiance that can be used for any purpose. The data accuracy is ensured by a regular quality control against information from in situ systems such as ground stations [35]. At the PV facility, the Mean Absolute Error (MAE) committed for the temperature with respect to NWP maps is $2.12{\;}^{o}C$. Likewise, the MAE obtained between the CAMS and the PV station is $46.97\;W/m^{2}$, for the whole year of measurements. This database is used to provide the forecasting models with a large GHI dataset for training purposes. Finally, the third category comprises non-stochastic data, such as sun position, used for the CCF calculation to determine the type of day; the extraterrestrial radiation for generating the forecasts and working out the irradiance on the tilted plane of the PV modules; and the installation parameters which are required for the PV power forecasting, as is explained in the following sections.

### 2.2. Data Preprocessing

The information obtained from the NWP-based weather forecasts must be transformed into numerical values. The forecasting time resolution is set to 15 min, mainly to follow the European Electricity Market Directive to be implemented in the coming years, which sets 15-minute energy matching periods. However, the AEMET only generates the weather maps hourly. This poses the inherent problem of merging time series with different time steps. For instance, for the PV power forecasting, the cell temperature (based on the ambient temperature) and the irradiance on the tilted plane are required. Since the latter has a time resolution of 15 min, so too should the time resolution of the time series for the cell temperature. To this end, quadratic interpolation is performed to create an oversampling of the NWP time series. Changes in the ambient temperature are usually smooth and it is assumed that the measurements shown in the NWP maps are defined with their intermediate values, since the Darboux property [36] is accomplished.

To prove the accuracy of this approach, Figure 2 depicts the ambient temperature obtained from the AEMET forecasts with respect to the values measured by a weather station located in the PV installation. The remarkable accuracy of the weather forecast for the temperature is noticeable. 

The CCF, on the other hand, is obtained by processing cloudiness information from weather maps. This parameter, which allows the type of the day to be defined, is used to identify those periods of time for which the presence of clouds can alter the PV power generation over a region through blocking the sun’s radiation. The CCF is obtained using a similar method as the work presented in [37], which provides a detailed description of how to calculate this parameter; mainly by detecting cloud-contaminated pixels in the weather maps that interfere between the sun and the installation. 

Finally, missing data can negatively affect the accuracy of the forecasts. To fill the missing gaps in the temperature and GHI datasets obtained from the weather station in the PV installation, GHI satellite data and the data from the NWP-based weather forecasts are used. Figure 3 shows an example of the reconstruction of missing data for the temperature and irradiance time series.

### 2.3. Model Design and Irradiance Forecasting

The third part in the forecasting framework deals with the LSTM-RNN-based model design and the forecasting itself, which aims to: (a) predict the mean PV power for a particular day with a 15-minute time step at the experimental PV facility, and (b) compute prediction intervals intended to show the likely uncertainty in the forecasting outcome [17]. This information constitutes an important input for the EMS in the VPP.

Figure 4 shows the flowchart of the model design and forecasting. The forecasting process starts with the LSTM-RNN model definition based on an iterative approach. Five years of GHI measurements from the Copernicus databases are utilized in the training process. The LSTM-RNN architecture depends on the characteristics of the input and output data and the cross-validation process. When creating the LSTM-RNN, 10% of the training set is used as the cross-validation set, optimizing the number of hidden layer units, mini-batch sizes, regularization factors, learn rate, and epochs (Table 1). Once these parameters are defined, the algorithm is extended to be used for future forecasts. The error in the training process is minimized by computing the RMSE, taking into account not only the proper convergence of the system but the computational time of the process. Squared errors lead the convergence in the LSTM-RNN as they are responsible for avoiding atypical errors, which have remarkable importance in energy management tasks. The architecture is composed of two input layers, one recurrent hidden layer (based on fifty memory blocks), and one output layer (Table 1). The memory block includes one or more self-connected memory cells along with four multiplicative gates (input, output, update, and forget gates). These gates provide the mechanism whereby the information can be stored and accessed over long periods of time, thereby avoiding the vanishing and exploding gradient problem posed by the conventional RNNs [38], e.g., the activation of the cell can be delayed, providing that the input gate remains closed to new inputs which can later become available by opening the output gate. The purpose of LSTM-RNN is, therefore, to model long-range dependencies. When training with sequential data, Gated Recurrent Unit (GRU), LSTM-RNN, and the Convolutional Neural Network (CNN)-LSTM are predominant in the literature [16]. As for CNN-LSTM models, they ensure higher accuracies for predictions based on more features which significantly compromise the computational time. It is worth noting that only two variables are used in this work. In [39] the authors show that these deep learning techniques ensure a higher accuracy than conventional ANNs or Support Vector Machines (SVMs) in GHI short-term forecasting. Consequently, LSTM-RNNs are used in this paper for the forecasting process. LSTM-RNNs achieve remarkable forecast accuracy with different prediction intervals, on account of their ability to memorize long historical data and determine the optimal time lags for the time series. These features are fundamental in the context of irradiance forecasting since there is no previous knowledge of the relationship between forecasts and the length of the historical dataset.

Once the LSTM-RNN model has been devised, the GHI prediction is made, followed by the estimation of the effective irradiance on the tilted plane of the PV module. Firstly, the calculation of the effective irradiance uses information from the two components of irradiance in the horizontal plane (direct and diffuse, since the albedo is zero in this case), calculated as a function of the clarity index ($k_{th}$), to obtain the diffuse fraction ($k_{dh}$) [40]. Once this information is obtained, the conversion into the tilted plane is estimated with the diffuse irradiance [41] and the albedo:(1)$$albedo=r_{o}\;ghm_{0}\;\left(1-cos\beta \right)/2$$
where $r_{o}$ is the albedo coefficient, considering that a value of 0.2, $ghm_{0}$ is the GHI and β is the tilted angle of the panels. Finally, the effective irradiance is determined by considering angular [42] and spectral [43] losses for p-Si modules and a typical moderate dust degree of DT=0.97 for the installation.

The Osterwald’s model [22] is used to convert the effective irradiance into PV power:(2)$$P_{DC}=SF\;\eta_{DC}\;P_{peak}\frac{G_{panel}}{G_{STC}}\;\left(1+\delta P_{m}\left(T_{cell}-T_{cell,STC}\right)\right),$$
where $P_{DC}$ is the PV power forecasted; SF represents the shading losses due to the surroundings of the installation, determined in Section 3.2 for this particular case; $\eta_{DC}=0.927$ includes wiring losses, module tolerances and mismatch losses; $P_{peak}=2.97\;\mathrm{kW}$ is the peak power of the installation; $G_{panel}$ is the effective irradiance of the panels previously calculated; $G_{STC}=1\;\mathrm{kW}/m^{2}$ is the irradiance under Standard Test Conditions; (STC), $\delta P_{m}=-0.4\%{/}^{o}C$ is the temperature coefficient of the PV panels of the installation; $T_{cell}$ is the cell temperature; and $T_{cell,STC}$ is the cell temperature under STC.

The cell temperature can be determined with the following expression, assuming the wind speed is negligible, since it can be considered as a nonsignificant effect complex to model because the wind does not affect each panel in the facility equally: (3)$$T_{cell}=\frac{T_{cell,NOCT}-T_{amb,NOCT}}{G_{NOCT}}G_{panel}+T_{amb},$$
where $T_{cell,NOCT}=45{\;}^{o}C$ is the cell temperature under Normal Operating Cell Temperature (NOCT) conditions; $T_{amb,NOCT}=20{\;}^{o}C$ is the ambient temperature under NOCT conditions; $G_{NOCT}=0.8\;\mathrm{kW}/m^{2}$ is the irradiance under NOCT conditions; and $T_{amb}$ is the ambient temperature, obtained from NWP forecasts.

Then, with the historical dataset of PV power forecasts, it is possible to compute prediction intervals for new forecasts. A prediction interval is an interval estimate for an unknown future value [17] which can be regarded as a random variable at the time when the prediction is made. In this paper, statistical prediction intervals are employed based on the work presented in [44], considering a Laplacian distribution model for the error as a function of the lead time, the launch time, and the type of day. Figure 5 shows the intervals for a specified day with 90% confidence, providing additional, valuable information from the forecast. PV power generation strongly depends on the weather conditions, the latter varying according to the season. This greatly hinders the ability of the forecasting algorithms to deliver accurate predictions, causing some degree of uncertainty which should be evaluated. Prediction intervals constitute the tool that can be used to express the degree of uncertainty of point forecasts which add a given confidence level. Additional details about the definition of the intervals, such as group selection and accuracy, are further explained in Section 3.3.

## 3. Results

This section presents the results obtained by the proposed intra-day forecasting strategy for VPP, which is divided into different steps: (a) GHI forecasting for a real VPP node and for an emulated VPP; (b) PV power estimation from the GHI forecasting output; (c) the quantitative assessment of prediction intervals; and (d) VPP scheduling. Firstly, the results are validated for a real PV installation, which plays the role of a VPP node. The PV installation is located in the Polytechnic School, at the University of Alcala (Madrid). Secondly, the strategy is developed for an emulated VPP, by using several ground-based meteorological stations uniformly spread over the Community of Madrid [33]. In order to evaluate the effectiveness of the model, a performance comparison in terms of accuracy/error, with respect to other methods proposed in literature, is also performed.

### 3.1. LSTM-RNN-Based GHI Forecasting for a Real VPP Node

The LSTM-RNN-based GHI forecasting for the real VPP node is performed by using measurements of irradiance taken in the PV facility located at the Polytechnic School of the University of Alcala (Spain). The initial training dataset is based on a 5-year period of irradiance values obtained from the CAMS dataset, since RNNs require a large amount of data for the learning process and GHI measurements are scarce in new installations. However, the test dataset is based on real measurements taken during the period from 1 June 2020 to 31 May 2021. Therefore, a whole year of real GHI values under different seasonal weather conditions are used to assess the accuracy of the forecasting approach. With a resolution of 15 min, the forecasting process starts at sunrise and ends at sunset. Furthermore, a new prediction is launched every 15 min and the dataset of irradiance is then updated, which ensures the accuracy of the results obtained. The network is trained with new measurements every day, during the night, to yield the best results. The GHI forecasts are given as a function of both the launch time and the lead time, parameters which are further defined, with the aim of computing the prediction intervals.

As far as the error assessment is concerned, this work relies on two types of metrics: (i) scale-dependent metrics such as the MAE and the Root Mean Square Error (RMSE); (ii) percentage-error metrics, such as the relative Mean Absolute Error (rMAE); and (iii) the relative Root Mean Square Error (rRMSE). Absolute values provide information about the average forecasting whereas the quadratic values are more sensitive to outliers, the combined analysis of the two allows for a thorough study of the results.. Error percentage values, on the other hand, provide an intuitive understanding of the error committed, which allows for a fair comparison to be conducted since the dependence on the magnitude is removed. However, when these values are near zero, scale-dependent metrics constitute the preferred option. The error metrics are summarized in Table 2, where $Y_{t}$ is the measured data at time t, $\hat{Y_{t}}$ is the forecast value at time t, and T is the length of the time series used to assess the accuracy of the algorithm.

The value of $Y_{t}$ denotes GHI at a specific hour of the day, t, and ${\hat{Y}}_{t^{\prime },t}$ is the prediction of $Y_{t}$ at $t^{\prime }$. The initial time, $t_{0}$, is fixed for each day and corresponds to the sunrise. To assess the error, two parameters are defined: lead time and launch time. Lead time corresponds to $\left(t^{\prime }-t\right)$ and is the difference between the time instant of the prediction and the moment when the prediction is launched. Launch time, on the other hand, is denoted by $\left(t^{\prime }-t_{0}\right)$ and is the difference between the current time and sunrise. Launch and lead time for the predictions of a particular day are better explained in Figure 6. When the launch time is fixed and the lead time is used as a parameter, a vector of predictions is obtained. However, when both parameters are set to a value, a single point forecast is obtained (red diamond in Figure 6).

The 3D plot in Figure 7 depicts the errors as a function of the lead time and the launch time which leads to the following conclusions. Firstly, for the scaled error, a high error rate is observed for short launch times under medium lead times. It is expected that the scaled error is large under the previous conditions since the radiation is high. However, as the launch time increases, this error significantly decreases. Secondly, it was clear that the lower the radiation, the smaller the scaled error; however, for percentage errors, the opposite is the case; when the launch time is small (less than 1 h), the percentage error is high, irrespective of the lead time. These plots give some insight into the prediction behavior and become particularly useful in enhancing confidence in the prediction with respect to other forecasting techniques. In this particular case, the intra-daily prediction is used when the mean error is smaller than the day-ahead prediction [37]. Finally, prediction intervals are derived from the MAE, assuming a particular distribution and splitting the predictions into groups as a function of the lead time, the launch time and the type of day, being very useful when a high degree of accuracy is required for the prediction.

Finally, the predictions obtained by the LSTM-RNN used in this work are compared with those available in the literature, which are depicted in Table 3. It is worth noting that this comparative analysis should not be strictly considered, since each dataset can have a relative influence on the performance. Nevertheless, some preliminary conclusions can be drawn from the study. Firstly, taking into account other widely used techniques from [45], the forecast error obtained in this work, in terms of the rMAE, is much smaller under short lead times (15 min), increasing until a similar value of the error is obtained under large lead times (6 h). A good performance under small forecast horizons is also obtained when comparing the results with [46] for a statistical AutoRegressive Integrated Moving Average (ARIMA) model, in terms of the MAE, obtaining a similar error to that of traditional RNNs, and a higher error with respect to a similar LSTM-based approach presented in [46], despite considering other inputs highly correlated with the irradiance. Finally, comparing the strategy presented in this paper with respect to the deep learning techniques (GRU, LSTM-RNN, and CNN-LSTM) from [39,47,48,49], a similar performance can be observed. To conclude, for small lead times, the forecasting approach introduced in this paper yields better results than those obtained by traditional methods. However, the forecasting error of the proposed LSTM-RNN-based method increases for higher lead times, until a similar performance is obtained with respect to the traditional methods compared from the literature. It is also observed that an increase in the number of inputs seems to slightly improve the performance of the forecast approach. Adding exogenous inputs to the forecast process is an alternative which is often used by researchers but negatively affects the performance when those resources are not available.

### 3.2. PV Power Estimation from the Forecasted GHI

The following step consists of estimating the power delivered by the PV modules from the GHI forecasts. To this end, the following parameters are required: (i) the prediction time instant; (ii) the site location in terms of latitude, longitude, and altitude; (iii) the installation characteristics, which include the orientation and inclination of the panels, rated parameters of the PV models available in datasheets, and losses associated with each part of the installation; and (iv) the ambient temperature, obtained from NWP maps. As stated above, analytical techniques exist to achieve this goal and, as a result, it is possible to quantify the error committed in the procedure.

This section focuses on two different approaches Firstly, real measurements of PV power are compared against the estimated values of PV power obtained from real measurements of GHI at the site. Secondly, the PV power is estimated from the forecasted values of GHI, evaluating the errors associated with the whole process. The GHI conversion searches for a reduced value of the error to maintain a similar performance to that obtained in the previous section, using the errors to construct the prediction intervals (Section 3.3).

Figure 8 depicts the comparison between the measured values of PV power at the site with respect to the PV power estimation obtained from real GHI measurements taken at the site. Three types of days have been selected: a cloudy day, an overcast day and a sunny day. The *x* axis is expressed in solar time. It is worth noting that the experimental setup at the site location has a building near the PV panels that generates partial shadows on some of them, starting from 16:36 and continuing until sunset. This event is also modelled in Equation (2), assuming a linear variation of this effect with respect to time (in Figure 8 SF=0.95 at 16:36, decreasing until SF=0.4 at sunset), and it also varies depending on the season of the year. Results show a reduced value for the error similar to that reported in other works [28], obtaining an rMAE=2.54% for sunny days, an rMAE=3.04% for partially cloudy days, and an increased value of rMAE=4.03% for overcast days. In terms of the squared error, values range from rRMSE=3.44% on sunny days and rRMSE=3.90% on partially cloudy days, to rRMSE=5.95% for overcast days. The transient characteristic of the inverter MPPT controller reveals that, in the presence of passing clouds, the inverter operating point becomes unstable. This is the reason why the error increases on these days. However, this does not pose any problem for the forecasting process since the time interval is 15 min, which considerably mitigates this negative effect.

Finally, Figure 9 depicts the forecast error in terms of the difference between the measured and estimated PV power as a function of the lead time and the launch time. The shapes of the figures are similar to the previous section, with similar percentage errors. Therefore, from the figure, the same conclusions reached by analyzing Figure 7 can be drawn: (i) the scaled error is high for short launch times and medium lead times but decreases significantly as the launch time increases; (ii) for launch times of less than an hour the percentage error is high, irrespective of the lead time; and (iii) the percentage error is high at lead times higher than approximately 7 h. The forecast error, which is dependent on the lead time and the launch time, is used to generate the prediction intervals in the following section.

### 3.3. Prediction Intervals of the Forecasted PV Power

Prediction intervals provide additional information about the plausible range of PV energy that will be generated at the site, for a defined confidence level selected by the user. Prediction intervals also indicate the degree of uncertainty in point forecasts. This could avoid unexpected energy shortages or, by contrast, energy surpluses, which are less critical than the former since the inverter can change its operational point to produce only the energy needed, despite wasting an exploitable energy resource.

In this paper, prediction intervals are obtained based on the work carried out in [44]. Previous results show how dependent the forecast accuracy is on the lead time and the launch time. This fact is used to split the dataset of predictions and create groups, assuming a specific distribution which is built based on the *MAE*. Therefore, each group is defined by selecting a launch time and a lead time, obtaining 365 samples per group, since a whole year is forecasted on this research. Figure 10 shows different error distributions for launch time values of 2, 4, and 6 h, and lead time values of 1, 2, and 3 h. In all of them, a Laplacian distribution is considered, similar to the work carried out in [37] but as a function of the CCF. Prediction intervals ($E_{15m}\pm p_{s}$) for each subset can be defined in terms of the *MAE* under this assumption: for a Laplacian distribution, a percentile p of probability (1−s) has an interval of $p_{s}=\pm MAE\cdot \ln\left(2s\right)$.

More detailed distributions can be determined provided that the selected groups are also created as a function of the CCF. However, by considering 10 groups as presented in [37], the number of samples of each group is not sufficient to create a proper error distribution. To overcome this drawback, the number of CCF groups is reduced to three, using the type of day classification criteria (e.g., sunny, cloudy, and overcast). The CCF parameter has an hourly resolution, its value is 0 when the sun is not covered by clouds and 1 when the sunlight is totally blocked. The type of day is classified evaluating the CCF during the daylight hours, with an hourly weighting of the amount of energy produced during the day. After that, the k-nearest neighbors (k-NN) method is used to form the groups, since it allows the dataset to be split in a simple way, offering an independent solution for each site in the VPP.

The assumption of a Laplacian distribution for each new selected subset carries an error that is necessary to quantify. The Prediction Interval Coverage Probability (*PICP*) [50], in Equation (4), indicates the percentage of predicted values that are inside the interval selected, and it must be close to the confidence level ($\gamma_{L}$). The confidence level selected in this research is $\gamma_{L}=80\%$, although this parameter can be modified depending on the operational risks that the site can handle: the higher the risks, the higher the benefits from the installation:(4)$$PICP=\frac{1}{T}\sum_{t=1}^{T}\epsilon_{t},\;\mathrm{where}\;\epsilon_{i}=\{\begin{array}{c} 1\;\mathrm{if}\;x_{i}\in \left[L_{i},\;U_{i}\right]\\ 0\;\mathrm{if}\;x_{i}\notin \left[L_{i},\;U_{i}\right] \end{array}\;.$$

Figure 11 depicts the absolute difference between the confidence level and the *PICP* for each type of day, being an effective method when this difference is close to zero. On sunny days, the *PICP* is close to the confidence level across the whole area, except for high lead times under small launch times where the difference increases. On cloudy days, the *PICP* is quite different from the confidence level during sunset. Nevertheless, the difference is acceptable in the rest of the area. In this case, the forecast has a lesser value during sunset since the energy produced is significantly reduced. Hence, prediction intervals also offer valuable information on cloudy days. Finally, for overcast days, the difference between the *PICP* and the confidence level increases with respect to sunny days, but the magnitude is acceptable and the prediction intervals are still valuable. To conclude, there are some zones with a high difference between the *PICP* and the confidence level. However, these scenarios correspond to small PV power measurements with bad forecasting performance (Figure 9). Therefore, prediction intervals are of little value for these points, since the strategy presented in this paper does not focus on those cases.

### 3.4. Evaluation of the GHI Forecasting for an Emulated VPP

The effectiveness of the whole forecasting process has been demonstrated for a single PV installation, which plays the role of a VPP node, along with its limitations with respect to the launch time and the lead time. The next step consists of assessing the algorithm performance for a set of PV facilities, forming a VPP. There are, however, no additional PV installations available in the study. Therefore, seven ground-based meteorological stations located in the Community of Madrid, apart from the PV facility at the university, are used to emulate the VPP nodes. Their locations are depicted in Figure 12. These ground-based stations are equipped with GHI sensors which allow the GHI forecasts to be generated. As for the power conversion, the characteristics of the PV installation from the university are used to obtain the power estimation for each emulated VPP node (peak power, $P_{peak}=2.97\;\mathrm{kW}$, temperature coefficient, $\delta P_{m}=-0.4\%{/}^{o}C$, and the performance of the equipment).

The same results as those shown in Figure 9 are used to quantify the accuracy of the prediction. However, in this case, the PV power forecast for each station is individually evaluated and the sum of power forecasts of the stations represents the PV power generated by the VPP, whose forecast error is depicted in Figure 13. By doing so, the PV power obtained at each station can be compared with respect to the total PV power forecasted. It can be observed that the scaled values of the error (MAE and RMSE) are higher than those in Figure 9. However, there is an 8-fold increase in the peak power with respect to a single facility. As a result, by looking at the relative values of the error (rMAE and rRMSE) it can be noted that the performance of the prediction increased for the VPP. The accuracy improvement of the PV power forecast can be expressed as the difference between the VPP forecast error and the sum of the error on each installation, dividing that value by the mean error committed on a single installation, obtaining a mean value of 12.37% with respect to the MAE, and 11.84% with respect to the RMSE. The shapes of the figures lead to identical conclusions to those reached by the analysis in Figure 9. Therefore, the prediction intervals maintain their potential value for error forecasting in the case of a VPP.

## 4. Discussion and Conclusions

The technical development of VPPs must be supported by EMSs, for which PV power forecasting is an essential part. By knowing the energy produced by each VPP node, usually based on renewable resources such as solar technologies, it is possible to optimize the expected profit generated by energy exchanges with the grid operator. However, it is difficult to obtain PV power forecasts when it is necessary to gather information from several nodes scattered throughout a wide area, especially when the input data, required for the predictions, incur costs. This research presents a way of accomplishing this objective, using an LSTM-RNN-based strategy to, firstly, forecast the GHI by using a dataset of irradiance values derived from satellite data freely obtained from the CAMS, and secondly, estimate the solar power by utilizing a PV model of the installation. The forecast is updated during the day to achieve the highest accuracy, and prediction intervals are estimated as a function of the *MAE*. This provides a useful framework to understand the behavior of each installation that composes the VPP.

The first results provided are related to the GHI forecast for the installation and are based on the lead time and the launch time, which allow zones with a reduced error and a high level of confidence to be created in the shape of prediction intervals which depend on the type of day. The GHI error, as a function of the lead time and the launch time, shows a low performance when the launch time is lower than 1.5 h, corresponding to sunrise. To avoid this, the forecasting process can begin at 1.5 h after sunrise; before this time, this research can rely on the day-ahead prediction made in [37] to obtain the irradiance forecast. To assess the accuracy of the intraday forecast, the results have been compared with those in the literature, achieving similar results to those obtained from deep learning algorithms and outperforming traditional techniques. The distinction between the lead time and the launch time means it possible to create better comparisons with respect to the literature, but also means it is difficult to summarize the research with only one value. The *MAE* committed, without considering the lead time and the launch time, is of $44.19\;W/m^{2}$, which is coherent with other studies.

Once the irradiance is forecasted, the conversion to PV power is analytically calculated, minimizing the error, which ranges from 2.54% to 4.03% in terms of the *rMAE* and from 3.44% to 5.95% in terms of the *rRMSE*. The error committed in this case is similar to the errors found in other articles [26,28]. The shapes of the error matrixes show similar results to those presented above. Therefore, similar conclusions can be drawn. The global *MAE* committed in this case is 137.21 W in a PV facility of $2.97{\;\mathrm{kW}}_{p}$.

Prediction intervals are selected once the PV power forecast is available, which allow a range of plausible values of point forecasts to be obtained. The method considers a Laplacian distribution of the error and distinguishes between the lead time, the launch time and the type of day, which is selected with a k-NN algorithm as a function of the CCF. To verify whether the boundaries maintain the associated level of confidence, the *PICP* is calculated, obtaining values close to the selected confidence level of $\gamma_{L}=80\%$. In this case, results reveal a noticeable difference between the *PICP* and the confidence level on cloudy days close to sunset. However, the predictions at those hours have minor importance. It can be concluded that the selected prediction intervals are of great relevance.

Finally, the PV power forecast is created, and the prediction intervals are selected for the PV facility so that conclusions under a VPP environment can be drawn. In this case, a real PV facility and seven ground-based weather stations in the Community of Madrid are selected to emulate the VPP, obtaining an improvement in the accuracy of 12.37% with respect to the *MAE*, and 11.84% with respect to the *RMSE*. Similar conclusions can be reached regarding the error as a function of the lead time and the launch time. Therefore, the whole strategy can be applied under different scenarios for launch times higher than 1.5 hours, relying on the day-ahead prediction prior to this. For this case, the error matrixes also indicate the best moments to obtain the predictions of the nodes, making it possible to increase the reliability of the VPP operation.

The major limitation of this study is related to the information of temperature and cloudiness freely obtained in Spain from NWP maps. In locations where this information is not available forecasts cannot be provided. Future works will focus on the application of this strategy along with a day-ahead time horizon strategy to schedule the operation of a VPP, creating a software that simplifies the process.

## Figures and Tables

**Figure 1 sensors-21-05648-f001:**
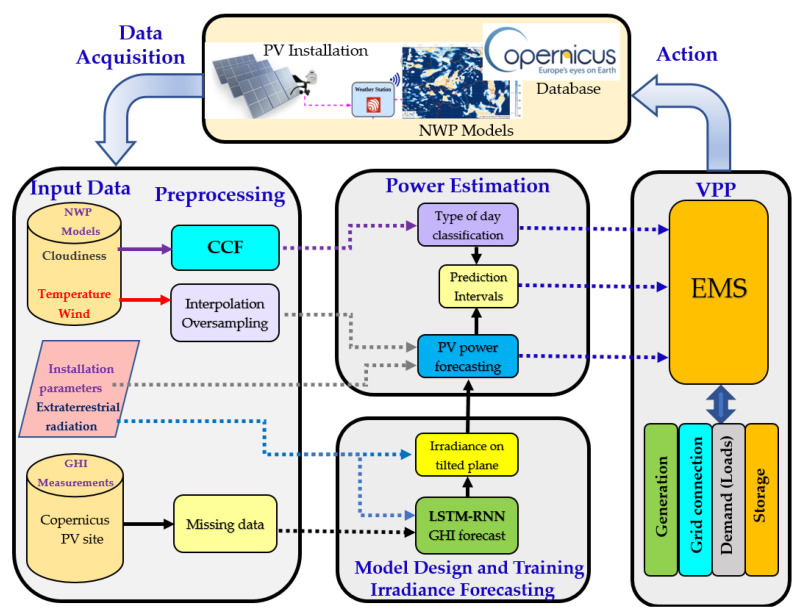
Forecasting framework.

**Figure 2 sensors-21-05648-f002:**
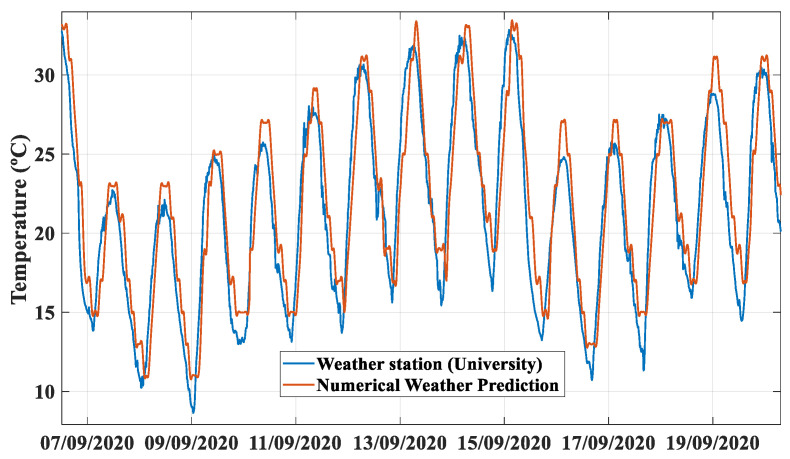
A comparison between the ambient temperature measured at the station and the temperature obtained from the AEMET website.

**Figure 3 sensors-21-05648-f003:**
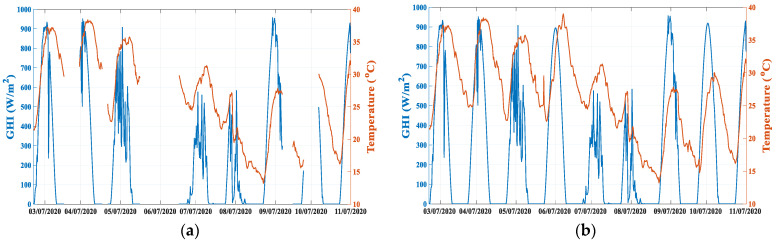
(**a**) Missing data of the GHI and the temperature on the site; (**b**) Time series reconstruction of the GHI and the temperature on the site.

**Figure 4 sensors-21-05648-f004:**
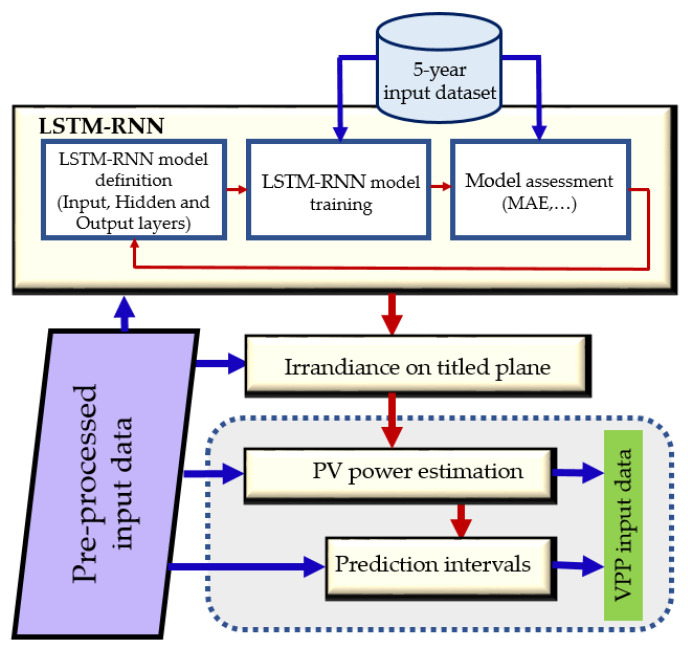
Flowchart for the LSTM-RNN-based forecasting model design.

**Figure 5 sensors-21-05648-f005:**
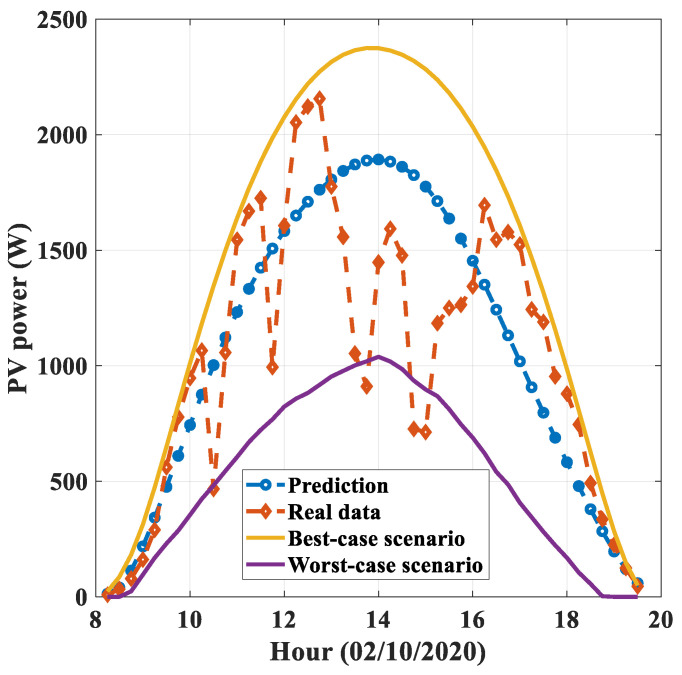
Prediction intervals with respect to the PV power.

**Figure 6 sensors-21-05648-f006:**
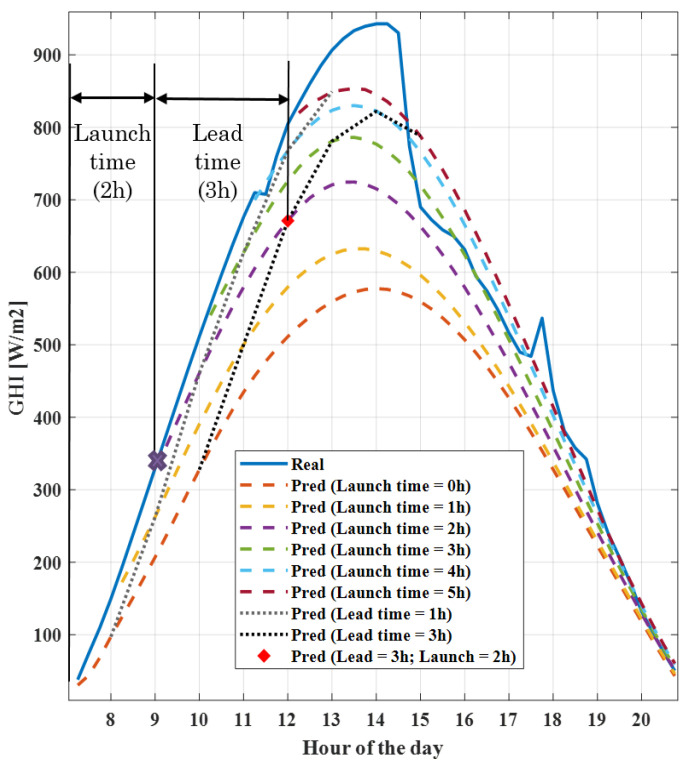
Real measurements of a selected day and its predictions. Dashed lines are the predictions for different launch times and dotted lines correspond to different lead times. Both parameters can be specified for a single day, obtaining the point forecast plotted with a red diamond.

**Figure 7 sensors-21-05648-f007:**
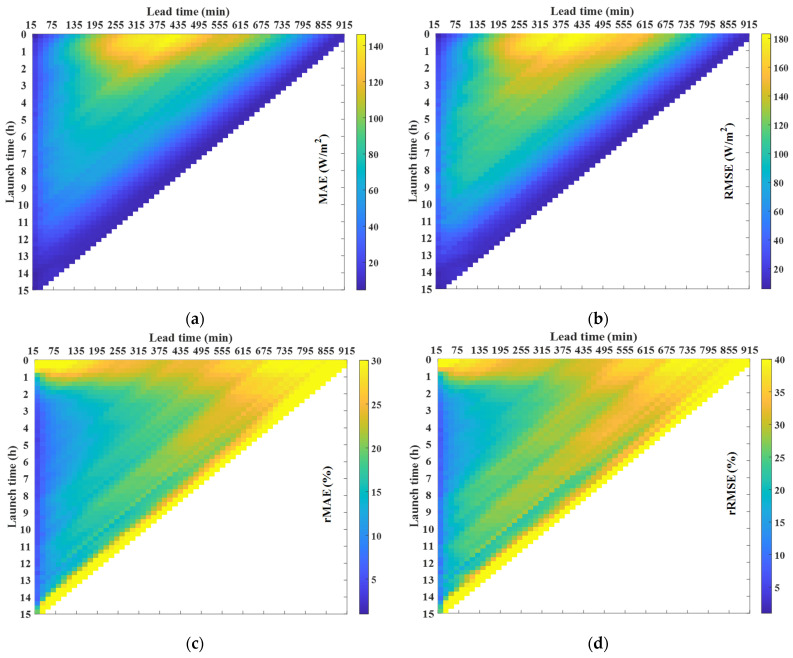
Error matrices obtained from GHI real measurements and GHI forecasted values, as a function of the launch time and the lead time: (**a**) MAE; (**b**) RMSE; (**c**) rMAE; and (**d**) rRMSE.

**Figure 8 sensors-21-05648-f008:**
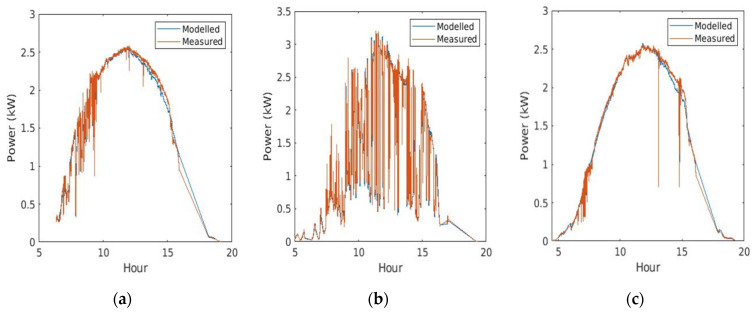
Comparison between the measured values of PV power with respect to values obtained from the conversion of real GHI measurements at the site. The selected days are: (**a**) a partially cloudy day (17 May 2021: rMAE=3.04% rRMSE=3.90%), (**b**) an overcast day (1 June 2021: rMAE=4.03% rRMSE=5.95%) and (**c**) a sunny day (4 June 2021: rMAE=2.54% rRMSE=3.44%).

**Figure 9 sensors-21-05648-f009:**
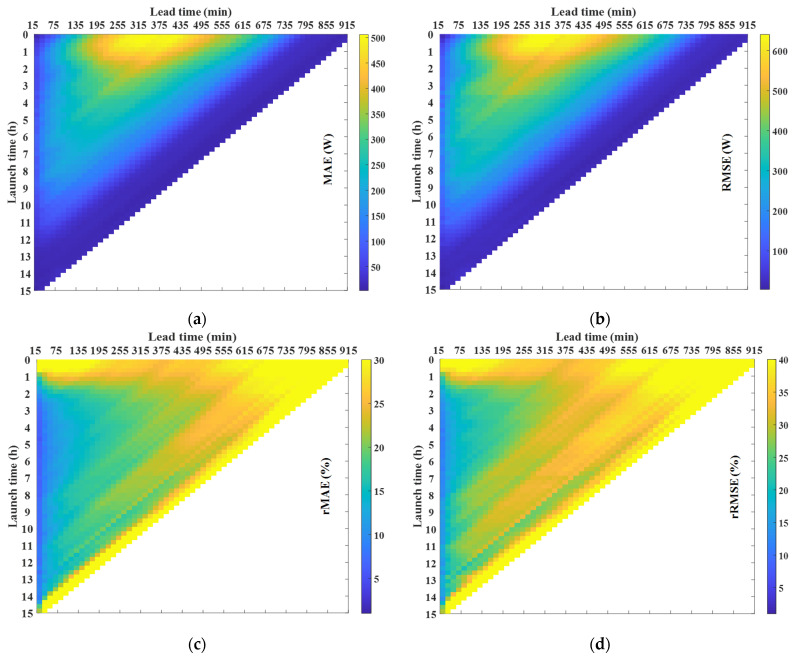
Error matrices obtained from PV power real measurements and PV power estimated values, as a function of the launch time and the lead time: (**a**) MAE; (**b**) RMSE; (**c**) rMAE; and (**d**) rRMSE.

**Figure 10 sensors-21-05648-f010:**
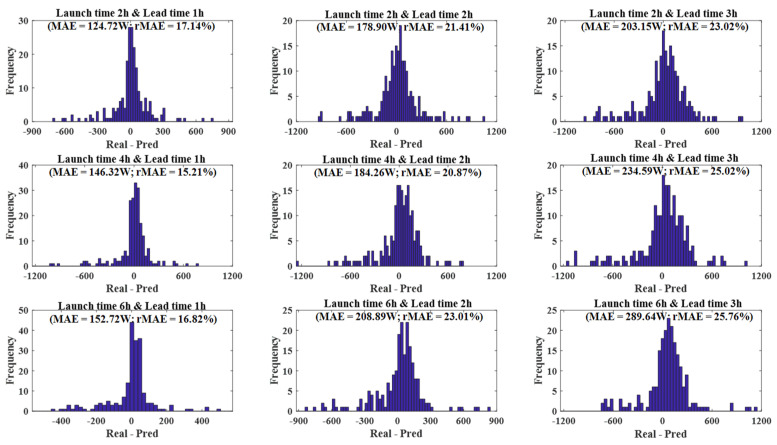
Error distribution for different subsets. A Laplacian distribution is assumed to create prediction intervals in terms of the MAE.

**Figure 11 sensors-21-05648-f011:**
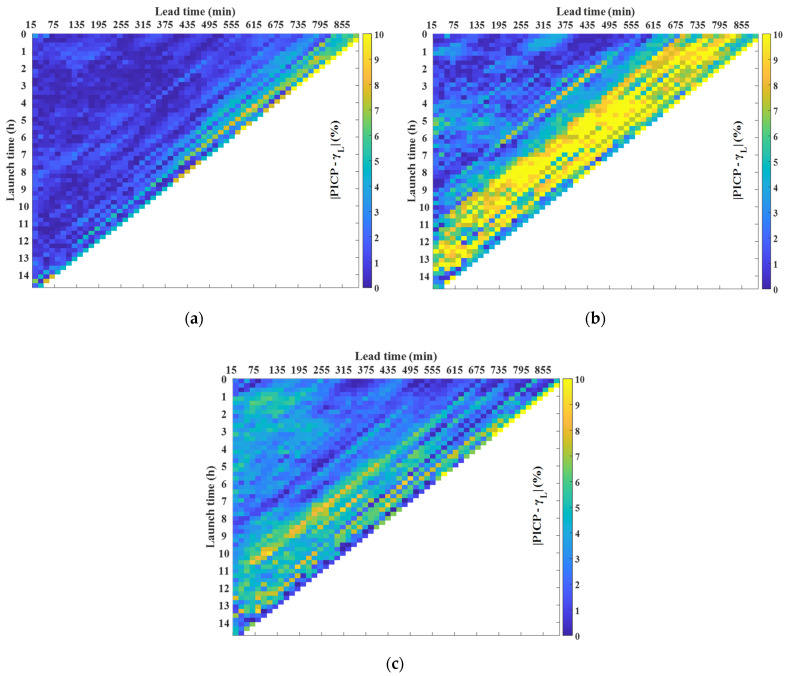
Absolute difference between the *PICP* and the confidence level for every subset selected on the prediction intervals for different types of day: (**a**) sunny; (**b**) cloudy; and (**c**) overcast.

**Figure 12 sensors-21-05648-f012:**
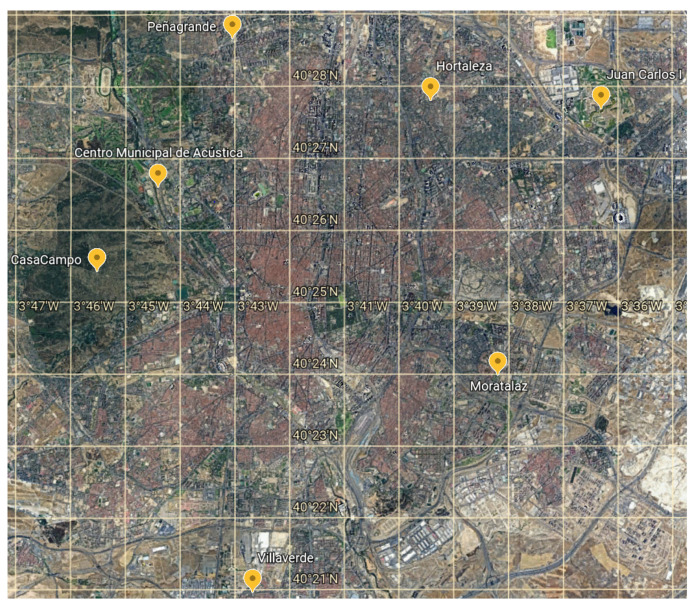
Location of the ground-based stations in the Community of Madrid used in the research.

**Figure 13 sensors-21-05648-f013:**
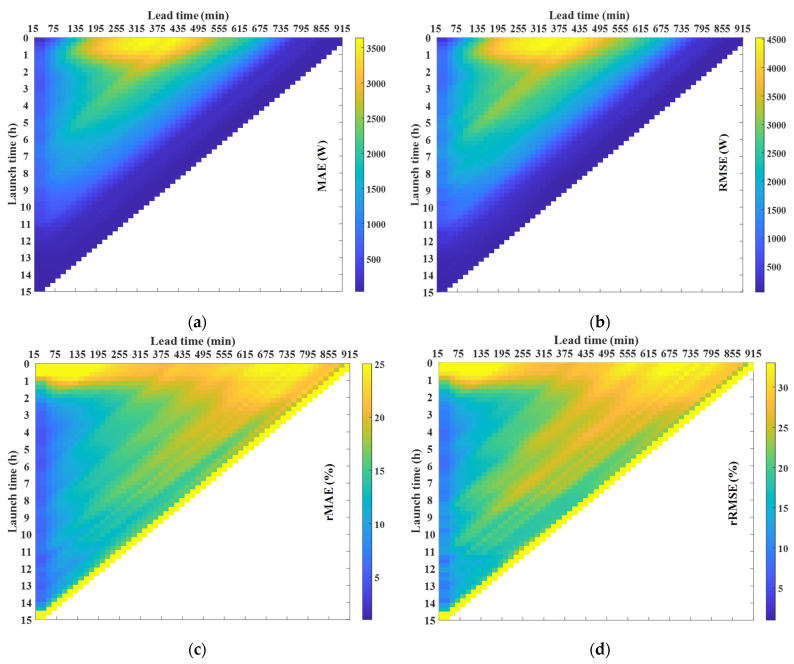
Error matrices of PV power forecasts from the VPP emulated in the research, represented as the sum of 8 different VPP nodes located in the Community of Madrid, as a function of the launch time and the lead time: (**a**) *MAE*; (**b**) *RMSE*; (**c**) *rMAE*; and (**d**) *rRMSE*.

**Table 1 sensors-21-05648-t001:** Parameters selected in the LSTM-RNN.

Number of Features	2 (GHI, Extra-Terrestrial Radiation)
Hidden layer units	50
Number of responses	1
Mini-batch size	256
Regularization factor	$5\times 10^{-4}$
Optimizer	Adam ($\beta_{1}=0.9,\;\beta_{2}=0.999,\;\epsilon =1\times 10^{-8}$)
Initial learn rate	0.01
Learn rate schedule	Piecewise (periodically)
Learning drop	0.5 every 20 epochs
Epochs	70
Limited gradient	1

**Table 2 sensors-21-05648-t002:** Metrics used to evaluate the model performance.

Metrics	Scaled (W/m^2^)	Percentage (%)
**Absolute**	$MAE=\frac{1}{T}\overset{T}{\underset{t=1}{\sum }}\left|Y_{t}-\hat{Y_{t}}\right|\;$	$rMAE=\frac{\frac{1}{T}\sum_{t=1}^{T}\left|Y_{t}-\hat{Y_{t}}\right|}{\frac{1}{T}\sum_{t=1}^{T}Y_{t}}\times 100\;$
**Quadratic**	$RMSE=\sqrt{\frac{1}{T}\overset{T}{\underset{t=1}{\sum }}{\left(Y_{t}-\hat{Y_{t}}\right)}^{2}}\;$	$rRMSE=\frac{\sqrt{\frac{1}{T}\sum_{t=1}^{T}{\left(Y_{t}-\hat{Y_{t}}\right)}^{2}}}{\frac{1}{T}\sum_{t=1}^{T}Y_{t}}\times 100$

**Table 3 sensors-21-05648-t003:** Comparison between the research results from this paper and those from other articles in the literature.

Model [Article]	Error	Forecast Horizon	Time Interval	Inputs	Results from This Paper
Smart pers. [45]	rMAE=(8–18)%	6 h	15 min	GHI, Clear Sky GHI, Cloud index maps, Cloud top height maps, …	rMAE=(4.17–17.73) %
CIAD Cast [45]	rMAE=(11–20)%				
Satellite [45]	rMAE=(10.5–19.5)%				
WRF-Solar [45]	rMAE=(12–18)%				
SVM-Radial [45]	rMAE=(7.5–15.5)%				
ARIMA [46]	$MAE=71.48\;W/m^{2}$	1 h	1 h	GHI, Clear Sky GHI, Cloud type, Temperature, Humidity, Precipitation, Wind, …	$MAE=41.88\;W/m^{2}$
RNN [46]	$MAE=41.83\;W/m^{2}$				
LSTM [46]	$MAE=31.86\;W/m^{2}$				
CNN-LSTM [39]	$MAE=41.88\;W/m^{2}$	1 h	1 h	GHI, Temperature, Wind, Precipitation, Humidity, Azimuth, …	$MAE=41.88\;W/m^{2}$
CNN-LSTM [39]	$RMSE=78.17\;W/m^{2}$				$RMSE=72.54\;W/m^{2}$
CNN-LSTM [39]	rMAE=10.58 %				rMAE=8.72 %
CNN-LSTM [39]	rRMSE=19.75 %				rRMSE=15.1 %
LSTM [47]	$RMSE=\left(77–143\right)\;W/m^{2}$	8 h	1 h	GHI, Humidity, Cloudiness, Temperature, Extra-terrestrial	$RMSE=\left(72–124\right)\;W/m^{2}$
LSTM [47]	rRMSE=(18.4–33)%				rRMSE=(15.1–29.2) %
GRU [48]	$RMSE=67.29\;W/m^{2}$	1 h	1 h	GHI, Zenith, Humidity, Temperature	$RMSE=72.54\;W/m^{2}$
LSTM [48]	$RMSE=66.57\;W/m^{2}$				
GRU [49]	$RMSE=58\;W/m^{2}$	30 min	1 min	GHI	$RMSE=55.78\;W/m^{2}$
LSTM [49]	$RMSE=55.29\;W/m^{2}$				

## Data Availability

The datasets generated for this study are available on request to the author Guillermo Moreno.

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
