# Peer review of "Intra-Day Solar Power Forecasting Strategy for Managing Virtual Power Plants"

_sensors, 2021, doi:10.3390/s21165648_

Round 1

Reviewer 1 Report

The article presents a detailed analysis of methods for forecasting solar irradiation and power generated by solar panels. Proposed own method based on LSTM-RNN.
The only drawback of the article is the insufficiently detailed description of some sub-processes used in the presented method, for example, the method for determining CCF and the expression for classifying the weather type of are not described.

Comment:
Why is the forecasting time resolution of 15 minutes chosen? How much the temperature forecasting error decreased / increased if select an interval of 1 hour?

Author Response

Dear reviewer, please see attached document.

Reviewer 2 Report

Dear Authors,

Thank You for the opportunity of reading this article.

General statements about the article:

-> The article discusses a method to efficiently produce intra-day accurate PV power forecasts at different locations, by using available free information. Thus the topic and scope of the article are interesting, actual, and highly desirable.

-> The article content suite to Sensors journal scope.

-> Keywords are correctly proposed.

-> Organization of the paper is clear and correct.

-> Literature review is based on 51. They are related to article content. A sufficient number of them are actual.

-> quality of the presentation is sufficient.

However, I indicated the following elements to revision:

#1

The abstract should be revised according to the journal's recommendations. The main findings should be additionally quantified and discussed.

#2

The results are well presented. But there is a luck of discussing them. Thus I recommend adding a separate section with discussion.

#3

Please extend conclusions. I suggest adding a paragraph with limitations of the proposed approach in the conclusions section.

Technical issues:

-> Please add required additional information: data availability statement

Author Response

(The authors gave the same response as above.)

Reviewer 3 Report

The article is valuable and very interesting. It does not contain major errors. However, it seems to me that the authors should introduce a few corrections. List of needed changes:

Rows 83, 84: There are no references. It should be fortified with some literature references.

The first part of the article lacks information where the measurement data comes from. There was also no information on what they were measured and what are the measurement errors. In order to fully illustrate the phenomena and compare the model with measurements, such information should be provided in the article.

Author Response

(The authors gave the same response as above.)

Reviewer 4 Report

This paper is well organized and of interest to power system researchers. The manuscript is mainly concentrated on the power forecasting of PV sites. Although PV sites are one the most important parts of VPPs, the title of the manuscript should be improved to convey solar power forecasting. The relevance to VPP is relatively low in the context. Please note that the power forecasting of a VPP is not limited to PVs only. 

Author Response

(The authors gave the same response as above.)

Round 2

Reviewer 2 Report

Dear Authors,

Thank You for the revision. All my comments were solved.

Thus I recommend to publish this article in present form .

Best regards,

Michał Jasiński